# Persistent Inflammation, Immunosuppression and Catabolism Syndrome (PICS) after Polytrauma: A Rare Syndrome with Major Consequences

**DOI:** 10.3390/jcm9010191

**Published:** 2020-01-10

**Authors:** Lillian Hesselink, Ruben J. Hoepelman, Roy Spijkerman, Mark C. H. de Groot, Karlijn J. P. van Wessem, Leo Koenderman, Luke P. H. Leenen, Falco Hietbrink

**Affiliations:** 1Department of Trauma Surgery, University Medical Center Utrecht, 3584 CX Utrecht, The Netherlands; rjhoepelman@gmail.com (R.J.H.); r.spijkerman-5@umcutrecht.nl (R.S.); kwessem@umcutrecht.nl (K.J.P.v.W.); l.p.h.leenen@umcutrecht.nl (L.P.H.L.); f.hietbrink@umcutrecht.nl (F.H.); 2Center for Translational Immunology, Wilhelmina Children’s Hospital, University Medical Center Utrecht, 3584 CX Utrecht, The Netherlands; l.koenderman@umcutrecht.nl; 3Department of Clinical Chemistry and Hematology, University Medical Center Utrecht, 3584 CX Utrecht, The Netherlands; M.C.H.deGroot-3@umcutrecht.nl; 4Department of Respiratory Medicine, University Medical Center Utrecht, 3584 CX Utrecht, The Netherlands

**Keywords:** PICS, trauma, immunology, infectious complications

## Abstract

Nowadays, more trauma patients develop chronic critical illness (CCI), a state characterized by prolonged intensive care. Some of these CCI patients have disproportional difficulties to recover and suffer from recurrent infections, a syndrome described as the persistent inflammation, immunosuppression and catabolism syndrome (PICS). A total of 78 trauma patients with an ICU stay of ≥14 days (CCI patients) between 2007 and 2017 were retrospectively included. Within this group, PICS patients were identified through two ways: (1) their clinical course (≥3 infectious complications) and (2) by laboratory markers suggested in the literature (C-reactive protein (CRP) and lymphocytes), both in combination with evidence of increased catabolism. The incidence of PICS was 4.7 per 1000 multitrauma patients. The sensitivity and specificity of the laboratory markers was 44% and 73%, respectively. PICS patients had a longer hospital stay (median 83 vs. 40, *p* < 0.001) and required significantly more surgical interventions (median 13 vs. 3, *p* = 0.003) than other CCI patients. Thirteen PICS patients developed sepsis (72%) and 12 (67%) were readmitted at least once due to an infection. In conclusion, patients who develop PICS experience recurrent infectious complications that lead to prolonged hospitalization, many surgical procedures and frequent readmissions. Therefore, PICS forms a substantial burden on the patient and the hospital, despite its low incidence.

## 1. Introduction

In recent decades, there has been much improvement in the clinical care for trauma patients, which has led to a decrease in mortality by exsanguination and life-threatening inflammatory complications, such as Acute Respiratory Distress Syndrome (ARDS), sepsis and multiple organ dysfunction syndrome (MODS) [1,2]. In particular, the initial approach and resuscitation, the introduction of standardized operating procedures, early hemorrhage control and improved clinical critical care have contributed to this decrease [3,4,5,6]. 

The increased survival directly upon trauma and the fact that fewer patients die of inflammatory complications nowadays has led to an increase in average intensive care unit (ICU) stay [7]. A prolonged ICU stay is also referred to as a state of chronic critical illness (CCI) [8,9]. Although no consensus exists on the definition of CCI [10,11], >14 days ICU stay is often used in recent literature [8,9,12]. 

Some of these CCI patients suffer from poor wound healing, recurrent infections and a disproportionately slow recovery [7,8,13]. This syndrome has been described as the ‘persistent inflammation, immunosuppression and catabolism syndrome’ (PICS) [5,7,14], not to be confused with the post-intensive care syndrome [15]. PICS is characterized by chronic low-grade inflammation, suppressed host immunity and a loss of lean body mass, despite nutritional intervention [7,13]. It is not quite clear which patients develop PICS and what the clinical course of these patients is. As the course is suggested to be complicated with multiple infections and high mortality rates, it is essential to identify patients at risk for PICS as early as possible. Clinical risk factors described so far include a poor premorbid health status and an age of 65 years and above [8,13,16]. Also, previous studies investigated readily available laboratory markers to enable identification of PICS patients shortly after trauma. Markers suggested in the literature include decreased lymphocyte counts, increased elevated C-reactive protein (CRP) levels and substantial weight loss or persistent decreased albumin levels [5,7]. 

To date, perioperative care continues to improve, and the elderly population is growing. Although literature is scarce on the subject, it suggests that PICS will be an increasing problem for ICU patients [13]. Therefore, the objectives of this study were to investigate the incidence of PICS after trauma in a Level I trauma center to determine the clinical course of these patients and to test the postulated markers to identify PICS patients in this study population.

## 2. Materials and Methods

### 2.1. Study Design

We conducted a 11 year retrospective cohort study of severely injured trauma patients admitted to the University Medical Center Utrecht (UMCU) from 1 January 2007 to 31 December 2017. PICS patients were identified in two ways: (1) by their clinical course and (2) by markers described in the literature [5,7,16]. The incidence of PICS was calculated, and the sensitivity and specificity of the markers were analyzed.

### 2.2. Patients

As an intensive care stay of >2 weeks is the common denominator in most definitions of CCI and PICS, all patients ≥16 years of age with an ICU stay of 14 or more consecutive days during hospitalization were selected from the trauma registry database of the UMC Utrecht. Patients were excluded if they were admitted to the ICU for other reasons than critical illness (e.g., logistical reasons or requiring ventilation support due to spinal cord lesions). Also, patients with isolated neurotrauma (e.g., traumatic brain injury or spinal cord injury) were excluded because these patients often have a prolonged ICU stay in the absence of systemic pathologies. Isolated neurotrauma was defined as having brain injury without injuries in other regions with an abbreviated injury scale (AIS) ≤2 [17]. Patient characteristics and data concerning trauma mechanism, treatment and complications were obtained. These data were collected from the prospective trauma registry and supplemented with hematological data from the Utrecht Patient Oriented Database (UPOD) and clinical data from the medical record system. The technical details of the UPOD have been described elsewhere [18]. In short, this database is an infrastructure of relational databases that allows (semi) automated transfer, processing and storage of data, including administrative information, medical and surgical procedures, medication orders, and laboratory test results for all clinically admitted patients and patients attending the outpatient clinic of the UMC Utrecht since 2004.

### 2.3. Definitions, Variables and Outcomes

CCI is used for patients with a prolonged ICU stay and PICS is used for a subset of CCI patients who exhibit a clinical phenotype consisting of inflammation, immunosuppression and catabolism. However, no uniform definitions exist for either CCI or PICS. An overview of the definitions used in this study is given in Figure 1. We defined CCI as an ICU stay of ≥14 days. Furthermore, we defined PICS clinically as: ICU stay of ≥14 days, ≥3 infectious complications and evidence of a catabolic state. A catabolic state was defined as either weight loss >10%, Body Mass Index (BMI) <18 or albumin <30 g/L during hospitalization [7]. If none of these markers for catabolic state were available (which was the case in 4/78 patients), the medical record system was searched for the nutritional state of the patient in dietitians’ notes, explicitly stating if the patient was in a catabolic state or not. An infectious complication was defined as an infection during hospitalization which required some kind of intervention (either pharmacological, surgical or radiological) or a readmission due to an infection after discharge. Patients meeting these criteria were considered to be the “clinical PICS group”. This group was compared to other CCI patients not meeting these criteria. 

Secondly, an attempt was made to identify PICS using the markers suggested in the literature (the PICS markers-positive group) [5,7]. Patients were considered to be PICS markers-positive if they had ≥2 days of immunosuppression (lymphocyte count <0.8 × 10^9^/L), ≥2 days of inflammation (CRP >50 mg/L) and a catabolic state as described above, during the first 30 days of hospitalization. PICS markers-positive patients were compared to other CCI patients not meeting these criteria (PICS markers-negative).

Infectious complications, hospital length of stay (LOS), ICU LOS, readmissions due to infectious complications, surgical procedures during hospitalization and surgical procedures after discharge, but related to the initial trauma or to acquired complications, were collected. Sepsis was defined as an increase in the sequential organ failure assessment (SOFA) score of 2 points or more in response to infection, according to the third international consensus definition for sepsis (Sepsis-3) [19]. Multitrauma was defined as Injury Severity Score (ISS) ≥ 16 [20]. Massive transfusion was defined as >10 units packed red blood cells within 24 h.

### 2.4. Statistical Analysis

All data were analyzed with IBM SPSS version 25(IBM Corporation, New York City, NY, USA). Distribution of normality was tested using the Shapiro–Wilk test. Continuous patient characteristics were presented as median with interquartile range (IQR) and compared using the Mann–Whitney U test, because the data were not normally distributed. The Fisher exact test was used to compare categorical patient data. Laboratory values are depicted as median with IQR because values were not normally distributed. Statistical significance was defined as a *p*-value < 0.05. Incidence was determined using the clinically identified PICS patients and calculated per 1000 trauma and multitrauma patients admitted to this hospital annually. The sensitivity and specificity of the markers were tested for our population using the clinical PICS group as gold standard. To compare lymphocytes and CRP between groups over time and to correlate for within-subject correlation, a linear generalized estimated equation (GEE) was used with an AR-1 correlation structure.

### 2.5. Ethics

A waiver was provided by the institutional medical ethics committee for this retrospective analysis of this prospectively collected database. In addition, in line with the academic hospital policy, an opt-out procedure was in place for use of patient data for research purposes. The process and storage of data were in accordance with privacy and ethics regulations. Data were analyzed anonymously. 

## 3. Results

### 3.1. PICS Incidence

Between 2007 and 2017, 13,576 trauma patients were admitted to the UMC Utrecht. Of these patients, 3859 were multitrauma patients and a total of 183 patients were admitted to the ICU for 14 days or longer (Figure 2). Two patients were excluded because they had no ICU indication (but needed ventilation support due to preexisting pathologies, which could not be provided elsewhere due to logistical reasons), and a total of 103 patients were excluded because of isolated neurotrauma. 

The remaining 78 patients fulfilled the criteria for CCI. The incidence of CCI in our trauma population was therefore 5.7 per 1000 trauma patients and 20 per 1000 multitrauma patients. Of the 78 CCI patients, 18 patients met the criteria for the “clinical PICS group” (Table 1) and 22 patients met the criteria for the “PICS markers-positive group” (Table 2). A total of eight patients fulfilled the criteria for both groups. Based on clinically identified PICS patients, the incidence of PICS was 1.3 per 1000 trauma patients and 4.7 per 1000 multitrauma patients were admitted to our center.

### 3.2. Clinical PICS Patients vs. Other CCI Patients

Clinical PICS patients received massive transfusion twice as often as the other CCI patients (67% vs. 33%, *p* = 0.015). Despite the long ICU stay in both groups, the clinical PICS patients were hospitalized twice as long as the other CCI patients (83 vs. 40 days, *p* < 0.001) (Table 1). ICU stay and duration of mechanical ventilation were also significantly longer for the clinical PICS group (30 vs. 19 days, *p* = 0.006 and 17 vs. 20 days, *p* = 0.001, respectively). Thirteen clinical PICS patients (72%) developed sepsis, while only 10 other CCI patients (17%) developed sepsis (*p* < 0.001). Also, PICS patients developed significantly more infections with multi-drug resistant organisms (*p* = 0.000). The number of readmissions due to infectious causes was significantly higher in the clinical PICS group (67% vs. 13%, *p* < 0.001). Surgical procedures during hospitalization were required for all (100%) clinical PICS patients and 61% needed additional surgical procedures after discharge related to the initial trauma or due to infectious complications. These percentages were significantly lower for the other CCI patients (73%, *p* = 0.016 and 22% *p* = 0.001). The number of surgical procedures per patient was higher in the PICS group too, both during hospitalization (eight vs. two, *p* < 0.001) as well as after discharge (three vs. 0, *p* = 0.006). During hospitalization, one clinical PICS patient died due to sepsis. Of the other CCI patients, three died due to irreversible neurological injuries and one due to sepsis during hospitalization. Within 3 years after trauma, an additional two clinical PICS patients died, one due to sepsis and one due to an unknown cause. Also, six other CCI patients died, one due to sepsis and five due to unknown causes.

### 3.3. PICS Markers-Positive Group vs. Other CCI Patients

Patients in the PICS markers-positive group had a significantly longer ICU stay than those not fulfilling the criteria (27 vs. 19 days, *p* < 0.001) (Table 2). Also, these patients more often required dialysis (five vs. 0, *p* = 0.001) and were significantly longer mechanically ventilated (27 versus 17 days, *p* = 0.000). Significantly more patients had infectious complications in the PICS markers-positive group (90% vs. 66%, *p* = 0.045) and there were more infectious complications per patient in the PICS markers-positive group (median of 2 vs. 0, *p* = 0.008). Sepsis also occurred more in the PICS markers-positive group (50% vs. 21%, *p* = 0.025). During hospitalization, three patients died in the PICS markers-positive group (two sepsis, one neurological injury) and two patients died in the PICS markers-negative group due to neurological injuries. Within 3 years after trauma, an additional two patients died in the PICS markers-positive group (one sepsis, one unknown) and six patients died in the PICS markers-negative group (one sepsis, five unknown).

### 3.4. Testing the Accuracy of PICS Markers

#### 3.4.1. Sensitivity and Specificity

The total population consisted of 78 patients. Eighteen PICS patients were clinically identified (clinical PICS). Of these patients, eight patients were also identified as PICS patients using the PICS markers. Therefore, of the 22 patients who were in the PICS markers-positive group, 14 did not have clinical evidence of PICS. This resulted in a sensitivity of 44% and a specificity of 77% in our study population (Table 3).

#### 3.4.2. Lymphocytes and CRP

No significant differences were found in lymphocyte count (β = 0.02, *p* = 0.999) and CRP levels (β = −0.12, *p* = 0.999) between patients who developed clinical PICS and patients who did not develop PICS. Median lymphocyte count was within criterion values (>0.8 × 10^9^/L) and median CRP levels were outside criterion values (>50 mg/L) for both groups during the first month after trauma (Figure 3). Although not significant, there was a trend towards lower lymphocyte counts in clinical PICS patients during the first week after trauma. When comparing clinical PICS patients with lymphocytopenia (≥2 days of lymphocyte count <0.8 × 10^9^/L) to clinical PICS patients without lymphocytopenia (Table 4), these patients did have a longer ICU stay (27 vs. 19, *p* = 0.001), longer duration of mechanical ventilation (35 vs. 21 days, *p* = 0.001), more infectious complications per patient (six vs. three, *p* = 0.020) and more surgical procedures per patient after initial discharge (0 vs. five, *p* = 0.037).

## 4. Discussion

In this study, we identified 18 PICS patients in eleven years, which translates to an incidence of 4.7 per 1000 multitrauma patients admitted to our level 1 trauma center annually. Both in-hospital mortality and 3 year mortality was low in the clinical PICS group. It may seem that PICS is a negligible problem with rather low incidence rates and high survival rates. However, PICS forms a substantial burden on the daily life of patients who develop the syndrome. These patients often require frequent and complex medical care up to 3 years after trauma. Also, they experience recurrent, sometimes life-threatening, recognition infections and require recurrent surgical procedures for this.

Costs of long-term medical care, and especially critical and surgical care, are high. The costs of an ICU and ward bed were recently calculated at approximately €2200 and €450 per day in the Netherlands, respectively [21]. Since PICS patients had an average of 30 days in the ICU and 83 days in the hospital, the average cost of total hospital stay is estimated at €89.850 per PICS patient. Compared to the average hospitalized trauma patient, these expanses are estimated to be a thirteen fold difference at least, based on average Dutch length of hospital and ICU stay reported in the literature [22,23]. This is a gross underestimation, since it does not include costs of imaging, medication, surgical procedures [24], readmissions and long-term care facilities. Therefore, PICS accounts for a substantial part of trauma-related expenses and forms an exuberant financial burden on the hospital, despite its low incidence.

The adverse outcomes associated with PICS were most evident in the clinically identified PICS patients. Outcomes of patients identified by the previously suggested markers were not as bad as those of the clinical PICS patients, but slightly worse than outcomes of other CCI trauma patients. This suggests that the PICS markers are not accurate enough to detect the clinically relevant PICS patients with the worst outcomes. This was supported by the low marker sensitivity and specificity found in this study.

Besides these laboratory markers, which are described to facilitate the detection of PICS, an ICU length of stay of 14 days or longer is a frequently described criterion for PICS [8]. Therefore, we used this as an inclusion criterion in our study to make a selection in the many trauma patients who were admitted over eleven years. However, ICU care differs greatly among countries, making a cut-off in ICU days an arguable criterion. In some countries, patients are not admitted to the ICU because of critical illness, but because a higher nurse to patient ratio is needed for other reasons or simple monitoring is required. In other countries however, patients are only admitted to the ICU when they are severely ill or require mechanical ventilation, such as in our hospital [25,26]. When there is no need for mechanical ventilation, but patients do need more care than a ward can deliver, patients are typically admitted to the intermediate care unit (IMCU) in our hospital. Therefore, it is possible that higher PICS incidence rates are found in countries with no IMCU or with different admission criteria for the ICU.

Furthermore, retrospective identification of a catabolic state in trauma patients can be challenging. Although BMI < 18 is a fairly undisputed criterion for poor nutritional state, body weight can, especially in the ICU, fluctuate daily (e.g., under influence of fluid in and output) [27]. Albumin (half-life of 14–20 days) is widely used to assess the nutritional status of patients, but is also a negative acute-phase protein and thus decreases when a patient experiences physiological stress (e.g., infection, surgery or trauma) [28]. Other markers for catabolism also included pre-albumin, retinol-binding protein (RBP) and creatinine-height index (CHI) [7], but these were not retrospectively available. Pre-albumin is also a negative acute phase protein, but with a shorter (2–3 days) half-life and smaller body pool, which theoretically makes it a more reliable indicator for nutrition. However pre-albumin, RBP and CHI, are all greatly influenced by renal function, infection and trauma as well [28,29].

It is not surprising that the specificity of these markers was low. CRP is an acute-phase protein, that is elevated due to many causes, including trauma, surgical procedures and infections [30,31,32,33]. Many PICS and other CCI patients underwent surgical procedures and developed infections in their first week after admission. Furthermore, lymphocytes are affected by trauma and infection and failure to normalize lymphopenia was described to increase mortality after trauma [34,35]. However, no significant difference in lymphocyte counts was found between PICS patients and other CCI patients, suggesting that this marker is not adequate to identify PICS. On the other hand, PICS patients who did have low lymphocyte counts had worse outcomes than PICS patients who did not have this. Therefore, although lymphocyte counts alone cannot be used to identify PICS, lymphocyte counts might be a useful addition to identify the PICS patients with the worst outcomes.

Hence, the PICS markers described so far are nonspecific, insensitive and arguable, and there is a need for better biomarkers to identify PICS. Recently, automated flow cytometry became available which enables fast and point-of-care analysis of receptor expression of immune cells [36,37]. Further research should investigate if such analyses can be used to identify patients at risk for PICS in an early stage. Early identification of patients at risk would enable earlier interventions. Although no specific treatment options exist so far, these patients might benefit from an immunoprotective protocol. Nowadays, trauma patients at risk for adverse inflammatory outcomes, often undergo damage control surgery, damage control orthopedics and damage control resuscitation [38,39]. Patients at risk for PICS are likely to benefit from these damage control strategies as well. In addition, limited evidence suggests that enhanced nutritional support (e.g., the addition of addition of arginine and glutamine [40]) and adjusted prophylactic antibiotic strategies [41,42] can have an immunoprotective effect in critically ill trauma patients. It is tempting to speculate that combining these interventions into an immunoprotective protocol, could improve clinical outcomes of patients at risk for PICS. Furthermore, it is remarkable that the mortality rate amongst PICS patients is low, in contrast to what was previously suggested in the literature [43]. This suggests that it is justified to keep continuing supportive treatment in PICS patients despite recurrent severe adverse outcomes.

The main limitation of this study was that there is no consensus on the definition for PICS in the literature. PICS is considered a clinical diagnosis, often recognized by a disproportionally long ICU/hospital stay, recurrent and nosocomial infections, failure to rehabilitate and disproportional weight loss. Physicians identify PICS patients through a combination of these characteristics, which to date, have not been well translated into criteria. The PICS definition for this study (≥3 infections, ICU stay of ≥14 days, evidence of catabolic state) was therefore based on the clinical course of the patients and was chosen during a consensus meeting with trauma surgeons treating these patients. Although this was needed to objectify the incidence of PICS, it is possible that this led to a slight overestimation or underestimation of the PICS incidence. Another limitation was the retrospective design of the study. Due to this, we were only able to obtain regular laboratory data. Genomic data or phenotypic cellular data were not available. Furthermore, other measures (e.g., urine 3-methylhistidine) or modalities (e.g., dedicated ultrasound) to detect disproportional muscle loss, were not available. Also, a limitation of a retrospective design is the change of missing values. However, because the data were extracted from the UPOD, and the UPOD contains all hematological parameters irrespective of the requested parameter, the number of missing values (e.g., lymphocyte counts and CRP levels) was limited. Moreover, laboratory values are generally requested daily for trauma patients in the ICU. Therefore, there were barely any missing values during the patients’ ICU stay. 

In conclusion, there is a need for a clear PICS definition and better markers to detect PICS. Patients who develop PICS experience recurrent inflammatory complications that lead to frequent readmissions and surgical procedures. Furthermore, infectious complications are frequently the result of multi-drug-resistant organisms. Therefore, PICS forms a substantial burden on the patient and a significant burden on hospitals, despite its low incidence.

## Figures and Tables

**Figure 1 jcm-09-00191-f001:**
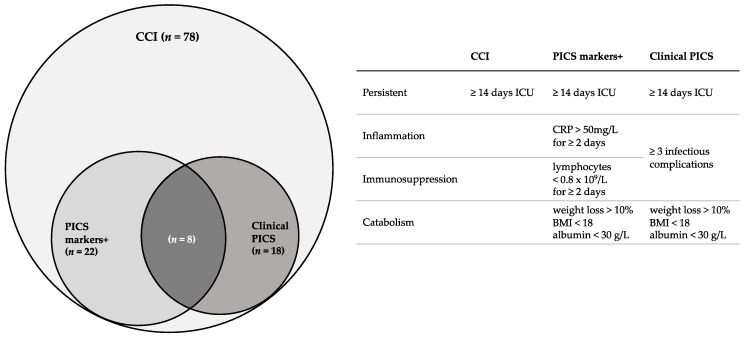
Overview of definitions used in this study and number of patients who fulfilled criteria according to these definitions. CCI = chronic critical illness, PICS = persistent inflammation, immunosuppression and catabolism syndrome, ICU = intensive care unit, BMI = Body Mass Index.

**Figure 2 jcm-09-00191-f002:**
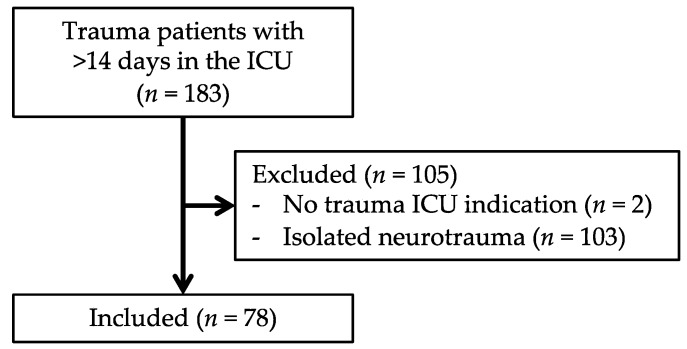
Flowchart of patient inclusion.

**Figure 3 jcm-09-00191-f003:**
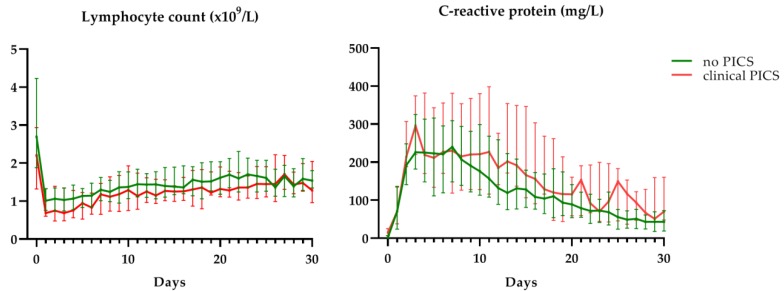
C-reactive protein levels and lymphocyte counts in clinically identified PICS patients (clinical PICS) in red and other CCI patients (no PICS) in green. Grey areas depict reference values. Data are presented as median with interquartile range.

**Table 1 jcm-09-00191-t001:** Patient characteristics of clinically identified PICS patients vs. other CCI patients.

Characteristics	Entire Cohort (*n* = 78)	No PICS (*n* = 60)	Clinical PICS (*n* = 18)	*p*-Value
Gender, male (%)	63 (80.8%)	48 (80.0%)	15 (83.3%)	0.999
Age	49 (32–65)	46 (28–62)	57 (45–68)	0.086
Injury severity score	34 (26–42)	34 (27–41)	29 (22–48)	0.548
Injury mechanism				0.261
Traffic accident	51 (65.4%)	39 (65.0%)	12 (66.7%)	
Fall from height	18 (23.1%)	15 (25.0%)	3 (16.7%)	
Crush injury	4 (5.1%)	2 (3.3%)	2 (11.1%)	
Other	5 (6.4%)	4 (6.9%)	1 (5.6%)	
Massive transfusion protocol	32 (41.0%)	20 (33.3%)	12 (66.7%)	**0.015**
Hospital days	43 (35–63)	40 (31–52)	83 (41–106)	**<0.001**
ICU days	20 (16–29)	19 (16–26)	30 (19–37)	**0.006**
Tracheostomy	45 (57.7%)	33 (55.0%)	12 (66.7%)	0.427
Mechanical ventilation days	20 (15–27)	17 (14–26)	20 (27–38)	**0.001**
Continuous veno-venous hemofiltration	5 (6.4%)	2 (3.3%)	3 (16.7%)	0.078
Infectious complications	57 (73.1%)	39 (65.0%)	18 (100%)	**0.002**
Per patient	1 (0–2)	1 (0–1)	3 (3–3)	**<0.001**
Sepsis *	23 (29.5%)	10 (16.7%)	13 (72.2%)	**<0.001**
Type of infections				
Pneumonia	40 (51.3%)	28 (46.7%)	12 (66.7%)	0.185
Surgical site infection	14 (17.9%)	6 (10.0%)	8 (44.4%)	**0.002**
Blood stream infection	15 (19.8%)	6 (10.0%)	9 (50.0%)	**0.001**
Infected OSM	7 (9.0%)	1 (1.7%)	6 (33.3%)	**<0.001**
UTI	5 (6.4%)	4 (6.7%)	1 (5.6%)	0.999
Abscess	4 (5.1%)	2 (3.3%)	2 (11.1%)	0.226
Other	12 (15.4%)	5 (8.3%)	7 (38.9%)	**0.005**
Multi-drug resistant organisms				**0.000**
MRSA	1 (1.3%)	0 (0%)	1 (5.6%)	
ESBL	5 (6.4%)	3 (5.0%)	2 (11.1%)	
MDR-GNB	5 (6.4%)	1 (1.7%)	4 (22.2%)	
Multi-drug resistant *Pseudomonas*	2 (2.6%)	0 (0%)	2 (11.1%)	
Multi-drug resistant *Acinobacter*	2 (2.6%)	1 (1.7%)	1 (5.6%)	
Infectious readmissions	20 (25.6%)	8 (13.3%)	12 (66.7%)	**<0.001**
Per patient	0 (0–0)	0 (0–0)	1 (0–3)	**<0.001**
Total infectious complications during or after hospitalization per patient	1 (0–3)	1 (0–2)	6 (3–7)	**<0.001**
Surgical procedures during hospitalization	62 (79.5%)	44 (73.3%)	18 (100%)	**0.016**
Per patient	3 (1–6)	2 (0–5)	8 (3–13)	**<0.001**
Surgical procedures after discharge	27 (34.6%)	16 (22.4%)	11 (61.1%)	**0.011**
Per patient	2 (0–2)	0 (0–1)	3 (0–5)	**0.006**
Total surgical procedures during or after hospitalization per patient	3 (1–7)	3 (1–5)	13 (2–22)	**0.003**
Discharge				0.931
Other hospital	8 (10.3%)	7 (11.7%)	1 (5.6%)	
Nursing home	11 (14.1%)	9 (15.0%)	2 (11.1%)	
Rehabilitation facility	27 (34.6%)	19 (31.7%)	8 (44.4%)	
Home (+/− additional care)	27 (34.6%)	21 (25.0%)	6 (33.3%)	
Total mortality	13 (16.6%)	10 (16.6%)	3 (16.7%)	1.000
In hospital	5 (6.4%)	4 (6.7%)	1 (5.6%)	1.000
<3 years	8 (10.3%)	6 (10.0%)	2 (11.1%)	1.000

All data are shown as n (%) or median (IQR). CCI = chronic critical illness. ICU = Intensive care unit. OSM = Osteosynthesis material. UTI = Urinary tract infection. MRSA = Methicillin-resistant *Staphylococcus aureus*. ESBL = Extended Spectrum Beta-Lactamase. MDR-GNB = Multidrug-resistant Gram-negative bacilli. Mortality <3 years does not include mortality during hospitalization. Variables are compared between patient meeting the clinical PICS criteria and those not meeting the clinical PICS criteria with a Fisher’s exact test, a Mann–Whitney U test or a Student *T*-test * Sepsis = defined as an increase in SOFA score >2 within a day in response to infection.

**Table 2 jcm-09-00191-t002:** Characteristics of CCI patients with positive PICS markers vs. other CCI patients.

Characteristics	Entire Cohort (*n* = 78)	No PICS Markers (*n* = 56)	PICS Markers (*n* = 22)	*p*-Value
Gender, male (%)	63 (80.8%)	45 (80.4%)	18 (81.8%)	0.999
Age	49 (32–65)	49 (33–66)	40 (27–63)	0.512
Injury severity score	34 (26–42)	34 (29–43)	29 (20–42)	0.147
Injury mechanism				0.702
Traffic accident	51 (65.4%)	36 (64.2%)	15 (68.2%)	
Fall from height	18 (23.1%)	14 (25%)	4 (18.2%)	
Crush injury	4 (5.1%)	3 (5.4%)	1 (4.6%)	
Other	5 (6.4%)	3 (5.4%)	2 (9.1%)	
Massive transfusion protocol	32 (41.0%)	21 (37.5%)	11 (50.0%)	0.322
Hospital days	43 (35–63)	42 (34–60)	50 (35–93)	0.281
ICU days	20 (16–29)	19 (16–25)	27 (23–36)	**<0.001**
Tracheostomy	45 (57.7%)	31 (55.4%)	14 (63.6%)	0.613
Mechanical ventilation days	20 (15–27)	17 (14–26)	27 (22–37)	**0.000**
Continuous veno-venous hemofiltration	5 (6.4%)	0 (0%)	5 (22.7%)	**0.001**
Infectious complications	57 (73.1%)	37 (66%)	20 (90%)	**0.045**
Per patient	1 (0–2)	1 (0–2)	2 (1–3)	**0.008**
Sepsis *	23 (29.5%)	12 (21.4%)	11 (50%)	**0.025**
Type of infections				
Pneumonia	40 (51.3%)	24 (42.9%)	16 (72.7%)	**0.024**
Surgical site infection	14 (17.9%)	11 (9.6%)	3 (13.6%)	0.746
Blood stream infection	15 (19.8%)	8 (14.3%)	7 (31.8%)	0.110
Infected OSM	7 (9.0%)	3 (5.4%)	4 (18.2%)	0.094
UTI	5 (6.4%)	4 (7.1%)	1 (4.5%)	0.999
Abscess	4 (5.1%)	1 (1.8%)	3 (13.6%)	0.066
Other	12 (15.4%)	7 (12.5%)	5 (22.7%)	0.303
Multi-drug resistant organisms				0.233
MRSA	1 (1.3%)	1 (1.8%)	0 (0%)	
ESBL	5 (6.4%)	3 (5.4%)	1 (4.5%)	
MDR-GNB	5 (6.4%)	2 (3.6%)	1 (4.5%)	
Multi-drug resistant *Pseudomonas*	2 (2.6%)	1 (1.8%)	1 (4.5%)	
Multi-drug resistant *Acinobacter*	2 (2.6%)	1 (1.8%)	1 (4.5%)	
Infectious readmissions	20 (25.6%)	13 (23.2%)	7 (31.8%)	0.565
Per patient	0 (0–0.25)	0 (0–0)	0 (0–1)	0.246
Total infectious complications during or after hospitalization per patient	1 (0–3)	1 (0–2)	2 (1–5)	**0.018**
Surgical procedures during hospitalization	62 (79.5%)	46 (82.1%)	16 (72.7%)	0.365
Per patient	3 (1–6)	3 (1–5)	3 (0–9)	0.720
Surgical procedures after discharge	27 (34.6%)	20 (35.7%)	7 (31.8%)	0.797
Per patient	2 (0–2)	0 (0–1)	0 (0–2)	0.937
Total surgical procedures during or after hospitalization per patient	3 (1–7)	3 (1–7)	3 (0–14)	0.980
Discharge				0.488
Other hospital	8 (10.3%)	5 (8.9%)	3 (13.6%)	
Nursing home	11 (14.1%)	9 (16.1%)	2 (9.1%)	
Rehabilitation facility	27 (34.6%)	20 (35.7%)	7 (31.8%)	
Home (+/− additional care)	27 (34.6%)	20 (35.7%)	7 (31.8%)	
Totally mortality	13 (16.6%)	8 (14.3%)	5 (22.7%)	0.500
In hospital	5 (6.4%)	2 (3.6%)	3 (13.6%)	0.133
<3 years	8 (10.3%)	6 (10.7%)	2 (9.1%)	1.000

All data are shown as n (%) or median (IQR). CCI = chronic critical illness. ICU = Intensive care unit. OSM = Osteosynthesis material. UTI = Urinary tract infection. MRSA = Methicillin-resistant *Staphylococcus aureus*. ESBL = Extended Spectrum Beta-Lactamase. MDR-GNB = Multidrug-resistant Gram-negative bacilli. Mortality <3 years does not include mortality during hospitalization. Variables are compared between patient meeting the clinical PICS criteria and those not meeting the clinical PICS criteria with a Fisher’s exact test, a Mann–Whitney U test or a Student *T*-test * Sepsis = defined as an increase in SOFA score >2 within a day in response to infection.

**Table 3 jcm-09-00191-t003:** PICS markers tested for sensitivity and specificity in the study population.

Total Cohort (*n* = 78)	Clinical PICS (*n* = 18)	Other CCI Patients (*n* = 60)	
PICS markers (*n* = 22)	*n* = 8	*n* = 14	Positive predictive value = 36%
No PICS markers (*n* = 56)	*n* = 10	*n* = 46	Negative predictive value = 82%
	Sensitivity = 44%	Specificity = 77%	

PICS markers+ = patients with positive PICS markers, PICS markers− = patients not fulfilling the PICS markers criteria. Sensitivity = 8/18 × 100. Specificity = 46/60 × 100. Positive predictive value = 8/22 × 100. Negative predictive value = 46/56 × 100.

**Table 4 jcm-09-00191-t004:** Outcome comparison of clinical PICS patients with lymphocytopenia ** versus clinical PICS patients without lymphocytopenia.

Characteristics	Normal Lymphocyte Count (*n* = 9)	Lymphocytopenia (*n* = 9)	*p*-Value
Hospital days	58 (40–128)	92 (71–110)	0.401
ICU days	19 (16–25)	27 (23–36)	**0.001**
Tracheostomy	4 (44.4%)	8 (88.9%)	0.131
Mechanical ventilation days	21 (16–27)	35 (31–60)	**0.001**
Continuous veno-venous hemofiltration	0 (0%)	3 (33.3%)	0.206
Infectious complications during hospitalization	9 (100%)	9 (100%)	0.999
Per patient	3 (3–3)	3 (3–4)	0.090
Multi-drug resistant organisms			0.217
MRSA	1 (11.1%)	0 (0%)	
ESBL	2 (22.2%)	0 (0%)	
MDR-GNB	1 (11.1%)	3 (33.3%)	
Multi-drug resistant *Pseudomonas*	0 (0%)	2 (22.2%)	
Multi-drug resistant *Acinobacter*	0 (0%)	1 (11.1%)	
Sepsis *	7 (77.8%)	6 (66.7%)	0.999
Infectious readmissions	5 (55.6%)	7 (77.8%)	0.620
Per patient	1 (0–1)	2 (1–5)	0.050
Total infectious complications during or after hospitalization per patient	3 (3–4)	6 (4–8)	**0.020**
Surgical procedures during hospitalization	9 (100%)	9 (100%)	0.999
Per patient	4 (2–10)	9 (6–15)	0.156
Surgical procedures after discharge	4 (44.4%)	7 (77.8%)	0.335
Per patient	0 (0–3)	5 (1–15)	**0.037**
Total surgical procedures during or after hospitalization per patient	7 (2–12)	17 (8–24)	0.051
Discharge			0.091
Other hospital	1 (11.1%)	1 (11.1%)	
Nursing home	1 (11.1%)	1 (11.1%)	
Rehabilitation facility	2 (22.2%)	6 (66.7%)	
Home (+/− additional care)	5 (55.6%)	1 (11.1%)	
Totally mortality	0 (0%)	2 (22.2%)	0.471
In hospital	0 (0%)	1 (11.1%)	0.999
<3 years	0 (0%)	1 (11.1%)	0.999

All data are shown as n (%) or median (IQR). ICU = Intensive care unit. OSM = Osteosynthesis material. UTI = Urinary tract infection. MRSA = Methicillin-resistant *Staphylococcus aureus*. ESBL = Extended Spectrum Beta-Lactamase. MDR-GNB = Multidrug-resistant Gram-negative bacilli. Mortality <3 years does not included mortality during hospitalization. Variables are compared between patient meeting the PICS markers criteria and those not meeting the PICS markers criteria with a Fisher’s exact test, a Mann–Whitney U test or a Student *T*-test * Sepsis = defined as an increase in SOFA score >2 in response to infection. ** Lymphocytopenia = defined as lymphocyte count below 0.8 × 10^9^/L for 2 or more consecutive days.

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
