# Peer review of "Persistent Inflammation, Immunosuppression and Catabolism Syndrome (PICS) after Polytrauma: A Rare Syndrome with Major Consequences"

_jcm, 2020, doi:10.3390/jcm9010191_

Round 1

Reviewer 1 Report

The authors' submit a succinct, well-written manuscript on an 1--year retrospective analysis seeking to identify the incidence of PICS among chronic critically ill patients after severe trauma.  The topic is salient and important to the readership as multiple reports have shown that while inpatient mortality continues to decline to historic levels, long-term morbidity, functional debilitation and mortality remains disturbingly high after ICU discharge.  Overall, I applaud the authors for pursuing this line of work, however, I have a few medthodologic concerns that need to be addressed, or at least more thoroughly explained.  I have several questions for the authors:

1)  The primary methodologic concern I have with this analysis is the exclusion of severely injured patients "Thirteen patients were excluded because they had no ICU indication (but needed ventilation support due to for example muscle weakness, which could not be provided elsewhere due to logistical reasons)".  The exclusion of these 13 patients is potentially concerning as prolonged ventilator support secondary to acute muscle wasting one of the primary clinical manifestations of PICS.  Excluding these patients potentially underestimates both the incidence and severity of the PICS population.  If this need for prolonged ventilation was secondary soley to neurologic issues such as spinal cord injury (severe TBI already excluded), than exclusion may be appropriate, but this needs to be clarified.  Otherwise the analysis should be repeated without excluding these patients.

2)  Again, more of a clarification of incidence numbers, only patients with high injury burden or hemorrhagic shock are at significant risk of developing PICS.  Previous reports show an incidence of CCI/PICS of approximately 20-30% in severely injured blunt trauma patients with evidence of hemorrhagic shock.  It is not surprising that the author's incidence is so low, when looking at all trauma admissions.  As PICS is thought to be instigated by the pro-inflammatory insult of hemorrhagic shock, what was the incidence of CCI/PICS among patients with hemorrhagic shock (pre-hospital/ED SBP<90 and/or Lactate>2)?  The incidence of CCI/PICS among patients in hemorrhagic shock should be reported (and emphasized) along with the existing data.  

3)  Can you report data on discharge disposition (i.e. home, rehab, another hospital, long-term care facility)?  Discharge to high-intensity care facilities rather than home or dedicated rehab in these patients has been found to be strongly associated with poor long-term outcomes.  

4)  The findings of the lack of sensitivity of previously reported PICS biomarkers is important, but dependent on their sampling.  These are not labs that are necessarily commonly drawn at regular intervals outside of research sampling.  Measuring catabolism and muscle loss is problematic by these critieria.  For instance, more recent reports use modalities such as dedicated ultrasound or CT imaging to detect muscle mass loss, or measures such as Urine 3-methylhistadine or measurements of the IGF/GH axis.  How much missing data was there, how was it dealt with, and could the rather crude initial definitions of PICS biomarkers and clinical definition be underestimating the incidence.  

Overall, this is an important paper and I look forward to the replies and revisions of the authors.

Reviewer 2 Report

The authors have explored patients admitted to their facility for more than a decade to determine the incidence of both CCI and PICS using an internally derived definition.  They correctly identify that there is no a single uniform definition to be used to identify each of these conditions.  Therefore, the incidence will directly vary with how each is defined.

The study is internally consistent and clearly presented.  What may be argued - and may benefit from being explored - is the perspective that all CCI patients also have PICS (since they have excluded those in the ICU for other reasons).  We are not presented with whether or not all of the CCI patients had non-resolving organ dysfunction, innate immune paralysis, T-cell exhaustion, and the like.  Form this retrospective database analysis, only some of those questions could be answered (organ dysfunction).  The decision to use > 3 infections seems high and was arrived at by consensus - but must have been a small number of individuals since the study was confined to one facility.

What would the incidence be if all CCI patients were included?  What if any other infection (> 1 infection) counted towards the definition?  Were any of the infections with MDRO? While we have data on lymphocyte count and CRP, there is no data on tracheostomy use, days of mechanical ventilation, dialysis use or days, MTP use, or reoperations in the same body cavity.  There is more that can be garnered from the data already collected that would inform the patient population determinations.  The definition used for sepsis seems to also include those with septic shock.  Should those patients be treated separately?

The authors suggest that we should identify the patient with PICS earlier.  What do they believe we should do once they are identified?  This merits its own paragraph.  The limitations section should also identify that we do not have genomic data, nor phenotypic data on those who suffered from sepsis. 
